# Parameters of Oxidative and Inflammatory Status in a Three-Month Observation of Patients with Acute Myocardial Infarction Undergoing Coronary Angioplasty—A Preliminary Study

**DOI:** 10.3390/medicina55090585

**Published:** 2019-09-13

**Authors:** Ewa Żurawska-Płaksej, Sylwia Płaczkowska, Lilla Pawlik-Sobecka, Hanna Czapor-Irzabek, Aneta Stachurska, Andrzej Mysiak, Tadeusz Sebzda, Jakub Gburek, Agnieszka Piwowar

**Affiliations:** 1Department of Pharmaceutical Biochemistry, Wroclaw Medical University, 50-556 Wroclaw, Poland; jakub.gburek@umed.wroc.pl; 2Diagnostics Laboratory for Teaching and Research, Wroclaw Medical University, 50-556 Wroclaw, Poland; sylwia.placzkowska@umed.wroc.pl; 3Department of Laboratory Diagnostics, Wroclaw Medical University, 50-556 Wroclaw, Poland; lilla.pawlik-sobecka@umed.wroc.pl; 4Department of Nervous System Diseases, Faculty of Health Sciences, Wroclaw Medical University, 51-618 Wroclaw, Poland; 5Laboratory of Elemental Analysis and Structural Research, Wroclaw Medical University, 50-556 Wroclaw, Poland; hanna.czapor-irzabek@umed.wroc.pl; 6Department and Clinic of Cardiology, Wroclaw Medical University, 50-556 Wroclaw, Poland; aneta.stachurska@gmail.com (A.S.); andrzej.mysiak@umed.wroc.pl (A.M.); 7Department and Clinic of Internal and Occupational Diseases and Hypertension, Wroclaw Medical University, 50-556 Wroclaw, Poland; 8Department of Pathophysiology, Wroclaw Medical University, 50-368 Wroclaw, Poland; tadeusz.sebzda@umed.wroc.pl; 9Department of Toxicology, Wroclaw Medical University, 50-556 Wroclaw, Poland; agnieszka.piwowar@umed.wroc.pl

**Keywords:** percutaneous coronary intervention, oxidative stress, presepsin, trimethylamine N-oxide, insulin-growth factor 1

## Abstract

*Background and Objectives*: Patients with acute myocardial infarction (MI) are usually treated with percutaneous transluminal coronary angioplasty (PTCA), which is burdened with a risk of postoperative complications, often accompanied by biochemical disturbances. The aim of our study was to evaluate a set of selected parameters of oxidative and inflammatory status, which could be useful in the management of post-procedural care in MI patients after PTCA. *Materials and Methods*: In this preliminary study, ischemia modified albumin (IMA), advanced oxidation protein products (AOPP), thiol groups (SH), total antioxidant status (TAS), insulin growth factor-1 (IGF-1), presepsin (PSP), and trimethylamine N-oxide (TMAO) were chosen as candidate biomarkers, and were determined in patients with MI who underwent PTCA at two time points: During cardiac episodes (at admission to the hospital, T0) and 3 months later (T3). *Results*: Most of the examined parameters were significantly different between patients and control subjects (except for IMA and TAS), but only hsCRP changed significantly during the time of observation (T0 vs. T3). Discriminant analysis created a model composed of AOPP, hsCRP, PSP, and TMAO, which differentiated male subjects into a group with MI and a control (without cardiovascular diseases). *Conclusion*: This set of parameters seems useful in evaluating inflammatory and oxidative status in MI patients after PTCA.

## 1. Introduction

Cardiovascular diseases (CVDs) are widespread in modern societies, constituting one of the leading causes of death all over the world. Their development is influenced by so-called cardiovascular risk factors, out of which lipid disorders, hypertension, diabetes, and tobacco smoking remain the most important [1,2]. It is also known that, apart from classical risk factors, many other factors, such as oxidative stress (OS) and chronic inflammation, have been implicated in the pathogenesis of CVD, and new facts about this complex clinical condition are still appearing [3,4]. Accordingly, there is a need for constant verification of the current state of knowledge. For example, although oxidative stress has been linked with CVD for many years, there are still no clinically useful guidelines indicating the best parameters of the oxidative–antioxidative balance [5]. The lipid oxidation products asymmetric dimethylarginine and glutathione S-transferase, and total antioxidant status (TAS), are the most frequently determined [6]. The concentration of thiol (SH) groups, advanced oxidation protein products (AOPPs), and ischemia modified albumin (IMA) are among the less frequently examined biomarkers [7]. Moreover, many inflammatory parameters, such as the high-sensitivity C-reactive protein (hsCRP), tumor necrosis factor alpha, or different interleukins (e.g., IL6), have also been evaluated as markers of the instability of atherosclerotic plaque, and they are potential tools for the early diagnosis of myocardial infarction (MI) and survival prediction [8]. However, only hsCRP is accepted as a clinically useful prognostic biomarker for CVD risk [9]. Recently, presepsin (PSP), generated in response to bacterial infection, was indicated as a promising novel marker for the diagnosis of MI [10,11]. Elevated presepsin levels is suggested as a predictor of infectious complications and consequent mortality in patients recovering from cardiac surgery [12]. Trimethylamine N-oxide (TMAO), another bacterial metabolite, has also been shown to be associated with the promotion of atherosclerosis, and increases the risk of major adverse cardiovascular events (MACEs) [13,14]. The role of insulin growth factor 1 (IGF-1) in CVD is still controversial. It was initially considered one of the factors in the development of atherosclerosis, due to the intensification of vascular proliferation and migration of smooth muscle cells, but some research indicates its cardioprotective role and association with lower mortality risk from CVD outcomes [15,16].

In modern cardiology, acute MI is routinely treated with percutaneous transluminal coronary angioplasty (PTCA). Large clinical trials show that this procedure enables a four-fold reduction in mortality and a three-fold reduction in the number of serious complications in patients with acute MI, as compared to pharmacological treatment only [17]. However, it is also burdened with a risk of postoperative complications, and induces disturbances in the biochemical balance of the organism, e.g., it exacerbates the inflammatory response, generates the production of free radicals, and also initiates the activation of many signaling pathways [18]. The aim of our preliminary study was to evaluate a set of markers to characterize the oxidative-inflammatory status in MI patients after coronary angioplasty, which might be important in the management of post-operative care. As candidate biomarkers, we chose: IMA, AOPP, SH, TAS, PSP, IGF-1, and TMAO. A panel of selected parameters was determined at two time points: During a cardiac episode (at admission to hospital) and 3 months later. To our knowledge, it has not previously been shown how these markers change over such a period of time or whether their measurement might provide some information about the state of MI patients after PTCA.

## 2. Patients and Methods

### 2.1. Study Population

The examined group consisted of 30 patients at the Department and Clinic of Cardiology of Wroclaw Medical University with first myocardial infarction (17 classified as STEMI, 13 as NSTEMI) assessed by standard clinical criteria, ECG monitoring, and elevated troponin levels, and qualified to PTCA according to Polish Cardiac Society guidelines [19,20]. Patients had mainly single-vessel disease (24 participants had 1-vessel disease and 4 had 2-vessel disease), and all of them received stents. Chronic kidney disease, chronic hepatitis, thyroid disorders, cancer, and infectious inflammatory states were applied as exclusion criteria. On discharge from the hospital, patients received lifestyle and pharmacological recommendations (angiotensin convertase inhibitors, beta-blockers, acetylsalicylic acid, clopidogrel). Moreover, the statin therapy that patients received prior to MI incidence was maintained. Early post-hospital rehabilitation included low-intensity resistance exercises 3 times a week, and individually matched general exercises. All the patients survived, and upon a control visit 3 months later they did not report angina symptoms. The control group (C) consisted of 30 healthy volunteers without a history of CVD or current inflammatory states, who declared that they were taking no medication. The study was approved by the bioethics commission of Wroclaw Medical University (number KB-47/2009, obtained 28.07.2017.) and informed consent was obtained from all participants.

Anthropometric measurements were made and routine laboratory parameters, such as morphology parameters, glucose concentration, lipid profile parameters, creatinine, and hsCRP concentration, were determined in all subjects. Moreover, patients had left ventricular (LV) mass, ejection fraction (EF), and the number and type of diseased vessels defined. Samples of venous blood were drawn at two time points: Upon hospital admission (T0) and upon a control visit 3 months afterwards (T3). Serum was obtained by standard centrifugation and stored at −80 °C until measurements.

SH groups, AOPP, IMA, and TAS were measured by colorimetric assays. SH group concentration was determined using 5-5′-dithio-bis(2-nitrobenzoic acid) at 412 nm with reduced glutathione as a standard [21]. AOPP concentration was measured using potassium iodide in the presence of acetic acid at 340 nm with chloramine T as a standard [22]. IMA level was estimated using cobalt chloride and dithiothreitol at 470 nm [23]. TAS was determined using 2,2′-azino-di-(3-ethylbenzthiazoline sulphonate) and potassium persulfate as an oxidant at 414 nm, with (*R*)-(+)-6-hydroxy-2,5,7,8-tetramethylchromane-2-carboxylic acid as a standard [24]. Moreover, the bromocresol purple dye-binding method was used for the estimation of serum albumin [25]. 

IGF-1, PSP, and insulin concentration were determined by immunoenzymatic methods using commercial ELISA kits: DiaMetra (Spello, Italy), Aviscera Bioscience (Santa Clara, CA, USA), and DRG International (Springfield Township, NJ, USA), respectively, according to the instructions provided by the manufacturers.

TMAO concentration was determined by the LC/MS/MS method developed in our laboratory. Briefly, 80 µL of deuterated TMAO (d_9_-TMAO), purchased from Cambridge Isotope Laboratories, USA (10 µM in acetonitrile:methanol 1:1 *v/v*) was added to 20 µL of serum sample, vortexed for 2 min, and centrifuged for 15 min (4 °C, 17,200× *g*). Supernatants were filtered with 0.22-µm nylon syringe filters (Chromacol, Cheshire, UK) before analysis. Simultaneously, standard solutions were prepared from TMAO purchased from Sigma-Aldrich, St. Louis, MI, USA and an internal standard of d_9_-TMAO was added to the standards in the same proportion (1:4). Next, a sample was injected onto a Luna 3-µm Silica 100 A column (Phenomenex, Torrance, CA, USA) and isocratic elution (acetonitrile:water 60:40 with 0.1% formic acid, 0.3 mL/min) was applied for compound separation (Dionex UltiMate 3000, Thermo Scientific, Waltham, MA, USA). Mass spectrometry detection (ESI-Q-TOF, Bruker Daltonics, Bremen, Germany) was carried out in positive ion mode with multiple reaction monitoring (MRM). The instrument parameters were as follows: Scan range: 30–600 m/*z*; drying gas: nitrogen; flow rate: 7.0 L/min; temperature: 200 °C; and potential between the spray needle and the orifice: 4.5 kV. Peaks were analyzed using Quant Analysis 2.2 software (Bruker Daltonics, Germany). The developed method was characterized by a good coefficient of variation (calculated as 1.77% for intra-assay CV and 6.58% for inter-assay CV), which indicated its satisfying repeatability.

### 2.2. Statistical Analyses

Data are presented as a median with a 25 to 75 interquartile range of observed values, and categorical variables are described by counts or percentages. The studied groups (T0, T3, and C) were compared by Mann U–Whitney or Kruskal–Wallis ANOVA test and T0 vs. T3 was checked by the Wilcoxon signed-rank test. Chi-squared test was performed for testing differences between categorical variables. Correlations between the parameters were examined with the Spearman test. Discriminate analysis was used for checking interrelationships between analyzed parameters of inflammatory and oxidative status and their contributions to differentiation between the studied groups. Statistical analysis was performed using Statistica PL version 13. A *p*-value < 0.05 was considered as significant.

## 3. Results

In Table 1, the anthropometric and biochemical characteristics of patients with myocardial infarction and control subjects are provided.

The studied groups did not differ in age, sex, or BMI value. The patients were characterized by significantly higher values of inflammatory, and some carbohydrate–lipid profile, parameters (WBC, hsCRP, glucose, TG, HDL) than controls. Serum creatinine reflecting renal function did not differ. Table 2 presents the results of measurements of all parameters of oxidative and inflammatory status in patients with myocardial infarction (T0), and after 3 months of recovery from PTCA (T3) in comparison with control subjects. The results of statistical analysis between all examined groups are also included in this table.

During myocardial infarction, SH groups, IGF-1, insulin, hsCRP, and PSP were significantly different when compared to the control group. After the 3-month follow-up, we observed statistically different levels of AOPP, SH groups, TMAO, insulin, and PSP in comparison to the control group. While most of the examined parameters were significantly different between patients (both in the T0 and T3 group) and control subjects, only hsCRP changed significantly during the observation time (T0 vs. T3). No statistically significant differences were observed for IMA or TAS. Furthermore, none of the examined parameters of inflammatory or oxidative status differed between all groups (T0, T3, and C). We also did not reveal any significant differences in the parameters of oxidative-inflammatory status when comparing patients with STEMI and NSTEMI (Appendix A). However, when comparing patients with normal (≥50%) and impaired (<50%) left ventricular (LV) ejection fractions, we observed significantly different levels of IMA, AOPP, and hsCRP in T0 and TMAO in T3 (Table 3). In patients with lower EF, a significant rate of changes (T0 vs. T3) was revealed for AOPP, IGF-1, and hsCRP, and for IMA and hsCRP in patients with preserved LV.

In the next step, we examined the potential relationship between these parameters in all groups. The statistically relevant results of the Spearman rank correlation analysis are presented in Table 4. The highest number of significant correlations was observed in the T0 group. 

Taking into account fact that male sex is one of the most important risk factors for MI, we conducted a discriminative analysis of oxidative-inflammatory status parameters in male subjects. A two-dimensional model was constructed on the basis of canonical discriminant analysis, in which IMA, AOPP, and SH groups, and TAS, IGF-1, PSP, TMAO, and hsCRP were included and forward step analysis was applied (Figure 1). A significant contribution to discriminant functions was revealed for the AOPP and SH groups, and PSP, TMAO, and hsCRP. Standardized canonical discriminant coefficients were used to rank the importance of the variables. In our study, the first discriminant function distinguished the mainly male control group from the male cardiac patients, and was the most weighted by AOPP, hsCRP, PSP, and TMAO, whereas the contribution of the second was relatively smaller. A similar analysis was performed for women, but due to the small size of the group, the results are not representative and have not been presented.

## 4. Discussion

Since, in patients with acute MI, percutaneous coronary intervention has become a routine treatment, secondary prevention, as well as cardiac rehabilitation, are essential parts of chronic management after revascularization [17]. The state of MI patients undergoing PTCA is not predictable and differs individually, since many additional physiological and pathological factors are involved. Like any medical procedure, PTCA is burdened by the risk of complications. The frequency of their occurrence depends on the severity and degree of changes in the arteries, accompanying diseases, general heart conditions (manifested mainly by EF), and age [26]. Full improvement of vascular functions after PTCA occurs progressively over a period of time. Hence, there is a need to monitor the patient’s status after intervention to accurately assess the efficacy of the procedure. Moreover, it is important to evaluate individualized postoperative patient states, which can be achieved through the assessment of biochemical markers. According to scientific reports, the majority of improvements after PTCA occur in the first 3 months [27]. This period also includes restoration of the disturbed biochemical balance, therein an amplification of proinflammatory mechanisms, coagulation imbalance, endothelial dysfunction, and oxidative–antioxidative disturbance [28]. The search for new markers that reflect disease regression after coronary intervention is still a challenge for contemporary cardiology. Therefore, we decided to examine some parameters, preselected on the basis of the latest literature data and our own experience, as potentially useful in characterizing the inflammatory and oxidative status of MI patients after PTCA. They represent a group of nonconventional risk factors for CVD, and their causal or effective relationship with MI has never been inspected. We evaluated how these parameters changed over a 3-month period of time, which is so far unknown.

IMA, AOPP, SH groups, and TAS, reflecting the intensity of oxidative stress, have been the subject of numerous studies on OS participation in the pathogenesis of many diseases, including CVD, neurodegenerative, digestive, and kidney diseases [29,30,31]. In the majority of these studies, the authors show statistically significant differences in the concentrations of various OS parameters between patients and healthy individuals. This confirms the contribution of OS to the development of these diseases, but on the other hand, suggests rather low specificity. Nevertheless, their potential usefulness in assessing patients’ state in CVD, especially when combined with other parameters, cannot be excluded [28,32]. Analyzing the obtained results, we observed that the concentrations of SH groups differed significantly between both patient groups (T0 and T3) and the controls, while in AOPP only T3 vs. C did (Table 2). At the time of infarction incidence (T0 group), SH group concentrations were 21% lower than in the control group and AOPP concentration was about 35% higher. These results confirm the intensification of OS during MI. However, we did not observe a significant rate of changes either for SH groups or AOPP concentrations during this short 3-month follow-up period. Some unexpected changes were, however, noted. AOPP concentration was the highest in the T3 group (approximately 18% higher than in T0), where oxidative status should have been improved. We suppose that this may have been caused by the long-lasting formation of the final oxidation products of proteins. Therefore, AOPP may be considered a late marker of OS in MI, which is additionally intensified by the procedure of coronary angioplasty [33]. Interestingly, we also observed the lowest IMA concentration during MI, which was almost at the same level as in the control group. This is contrary to the results of many studies showing the highest IMA levels about 10 min after the onset of ischemia, which should normalize within about 6 h [34]. In our study, after 3 months of recovery (T3 group), IMA levels increased by almost 12%. Perhaps this was induced by other non-cardiac factors, such as physical activity, which is currently recommended during cardiac rehabilitation. False negative IMA values are reported in the presence of lactic acidosis, which occurs during exercise, as well as during MI [34]. Nonetheless, based on analysis performed in subgroups of patients with normal and decreased LV EF (Table 3), we suggest that IMA may serve as an additional marker in the evaluation of patient states, but only in groups with the preserved function of the left ventricle (significant rate of changes T0 vs. T3). 

TMAO showed an ambiguous tendency. In the T3 group, it was only slightly higher (5%) than in the T0 group, and significantly lower (18%) in the control group than in patients. An elevated plasma TMAO level has previously been shown to be an independent risk factor of MACE (including myocardial infarction, stroke, need for revascularization, or death) over the ensuing 30-day and 6-month periods and its prognostic value was also assessed in patients who underwent coronary angiography in a one-year observation (OR 1.51) [35]. When used with the Global Registry of Acute Coronary Events (GRACE) score for calculating 6-month risk, TMAO showed its usefulness as a secondary risk stratification biomarker [14]. Referring to the obtained results, it is very likely that a 3-month period was not long enough to cause spectacular changes in TMAO levels. Exceptionally, in subgroups of patients divided according to the value of LV EF, we observed significant differences in its concentration after PTCA, which may indicate its analytical potential. Unfortunately, we did not perform investigations for longer than 3 months, hence we cannot draw conclusions about its prognostic importance, but incorporation of TMAO measurement into the diagnostic panel may provide some benefits in assessing MI patients’ state after PTCA. Our observations and the latest literature data indicate that it may become a new, unconventional CVD marker of great importance [36].

Due to the frequently occurring noninfectious inflammatory responses accompanying an extensive MI, we also investigated hsCRP and presepsin concentrations, as old and new parameters in CVD. We confirmed that the subclinical inflammation reflected by hsCRP concentration is significantly reduced after a 3-month follow-up, independently from left ventricular EF (Table 2 and Table 3). In the case of PSP, it was reported to increase (over twice) in patients with ST elevation in comparison to the control group and may have prognostic value for patient states after MI [11,37]. However, it is not known exactly how this parameter changes after coronary intervention. Our study is the first to evaluate serum PSP levels in MI patients over a 3-month period of observation, but we did not reveal any significant differences, even in the subgroup with STEMI type (Appendix A). Hitherto, only Saito et al. [38] have examined PSP level in patients undergoing cardiovascular surgery, and they reported the highest values on the first day after intervention, which may be associated with monocyte activation [39]. In our study, we observed a surprisingly lower concentration of PSP in cardiac patients than in the control group, even though the overall number of WBC was higher in patients. This may partly be explained by the possible anti-inflammatory effects of medicines taken (mainly statins and angiotensin convertase antagonists), but this requires verification [40]. Interestingly, we revealed a significant correlation between PSP and TMAO, both connected with bacterial metabolism, in T0 groups. In this group, TMAO also correlated with hsCRP, which together created the inflammatory network underlying MI.

From the other parameters examined in our study, IGF-1 is a factor connected with atherosclerosis development, but also participates in cardiac remodeling after MI [15,41]. Concentrations of IGF-1 were the lowest at the time of the infarction, and after three months, it had increased by 13%. In the control group, this parameter was 30% higher than in patients during the infarction. Yamaguchi et al. also found that serum IGF-1 levels were significantly lower in patients with MI and suggested that a reduced IGF-1 concentration was associated with a worse prognosis, reporting IGF-1 as an independent predictor of 90-day mortality [42]. Apart from its cardioprotective role, its vasculoprotective role is also considered, and this effect is provided by ameliorating endothelial dysfunction due to the stimulation of the production of nitric oxide in smooth muscle and endothelial cells [43]. The negative relationship between IGF-1 and IMA observed in our study, also indicated as a factor underlying subclinical vascular disease, remains in accordance with this hypothesis, but complex IGF-1 signaling in MI should be elucidated in further studies [44].

Although the above-described changes in levels of oxidative-inflammatory parameters cannot be directly linked with disease progression or recovery status, they may somehow reflect the biochemical status of patients with acute MI. It should be taken into account that the patients included in this study were basically in the same state—they all survived, and upon a control visit after 3 months, they did not complain of angina pain. However, we still observed significant changes in most of the examined parameters in comparison to the control group, and we believe that further research in this direction is worth continuing.

## 5. Conclusions

In conclusion, our results revealed that, among the examined parameters, AOPP, hsCRP, PSP, and TMAO constitute a promising marker set for evaluating inflammatory and oxidative status in MI patients after coronary intervention, especially in male subjects. Their changes are not influenced by infarction type, and their simultaneous measurement may provide some beneficial guidance in the decision-making process during post-operative care. Unfortunately, this is a single-institution observational study and we do not have sufficient data to draw bold conclusions. A relatively small patient population is one of the limitations of this study, which makes it especially difficult to interpret subgroup findings. Investigations should be continued with a larger study group and for a longer time of observation.

## Figures and Tables

**Figure 1 medicina-55-00585-f001:**
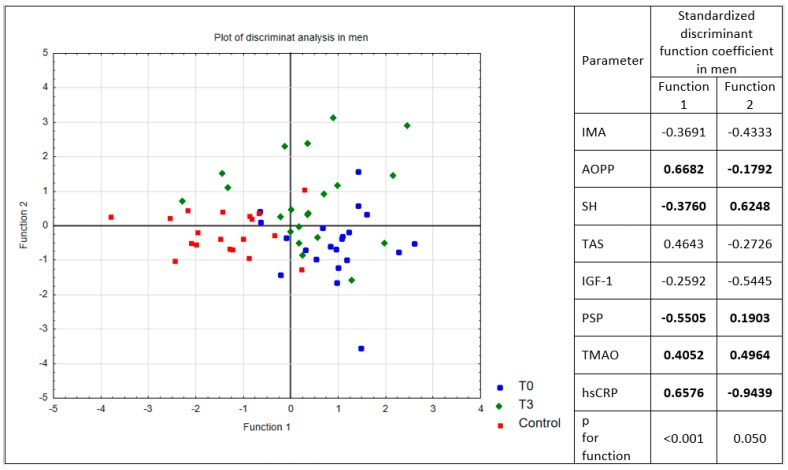
Scatter plot of canonical discriminant function one versus function two and values of standardized canonical discriminant coefficients in men, used to assign participants to T0, T3, and C groups. Figure legend: T0—patients during myocardial infarction, T3—patients after 3 months of recovery from percutaneous transluminal coronary angioplasty, C—control subjects, IMA—ischemia modified albumin, AOPP—advanced oxidation protein products, SH groups—thiol groups, TAS—total antioxidant status, TMAO—trimethylamine N-oxide, IGF-1—insulin growth factor-1, hsCRP—high sensitive C-reactive protein, PSP—presepsin.

**Table 1 medicina-55-00585-t001:** The anthropometric and biochemical characteristics of patients with myocardial infarction and control subjects. Values are given as median (25% and 75% percentile) or counts/percentage.

Parameter (Unit)	Patients (*n* = 30)	Controls (*n* = 30)	Statistical Significance of Differences [*p*-Value]
Age (years)	63 (54–75)	59.5 (56–63)	0.065
Male sex (*n*)	21	19	0.58
BMI (kg/m^2^)	27.1 (24.4–31.8)	25.05 (22.7–28.1)	0.057
SBP (mmHg)	120 (117–130)	130 (120–138)	0.104
DBP (mmHg)	72 (70–80)	82 (75–87)	0.051
WBC (10^9^/L)	8.68 (7.63–11.06)	5.65 (4.70–6.20)	**<0.001**
hsCRP (mg/L)	3.65 (2.26–9.90)	1.54 (0.84–2.96)	**0.001**
Glucose (mmol/L)	6.0 (5.4–7.0)	5.2 (5.0–5.5)	**<0.001**
Cholesterol (mmol/L)	5.4 (4.3–6.3)	5.9 (5.2–6.6)	0.075
TG (mmol/L)	1.51 (0.90–2.27)	1.11 (0.80–1.46)	**0.025**
HDL (mmol/L)	1.16 (0.98–1.34)	1.47 (1.21–1.65)	**0.002**
LDL (mmol/L)	3.38 (2.42–4.16)	3.49 (2.92–4.21)	0.53
Creatinine (µmol/L)	76.1 (62.8–92.1)	74.3 (64.6–90.2)	0.65
Troponin I (µg/L	7.1 (2.37–28.24)	NA	
LV mass (g)	315.5 (261–417)	NA	
EF (%)	60 (35–65)	NA	
Presence of DM (*n*)	6	0	0.031
Smoking habit (*n*)	13	9	0.423

Table legend: BMI—body mass index, SP—systolic blood pressure, DP—diastolic blood pressure, WBC—white blood cells, hsCRP—high sensitive C-reactive protein, TG—triglycerides, HDL—high density lipoprotein cholesterol, LDL—low density lipoprotein cholesterol, LV mass—left ventricular mass, EF—ejection fraction, DM—diabetes mellitus, NA—not applicable. Statistically significant differences are given in bold.

**Table 2 medicina-55-00585-t002:** Parameters (median with 25%–75%) of oxidative and inflammatory status in patients with myocardial infarction (T0) and after 3 months recovery from PTCA (T3) in comparison to control subjects.

Parameter (Unit)	Patients		Controls	Statistical Significance of Differences [*p*-Value]		
	T0	T3	C	T0 vs. T3	T0 vs. C	T3 vs. C
IMA (ABSU/g)	40.5 (34.5–51.8)	45.8 (39.8–57.3)	41.1 (38.6–49.1)	0.33	1.00	0.94
AOPP (μmol/L)	226 (106–389)	274 (180–391)	147 (130–183)	0.44	0.18	**0.002**
SH groups (mmol/L)	0.41 (0.35–0.43)	0.41 (0.23–0.46)	0.52 (0.45–0.57)	1.00	**<0.001**	**<0.001**
TAS (μmol/L)	17.2 (15.6–17.9)	16.6 (15.7–17.5)	16.3 (15.8–16.9)	1.00	0.11	0.54
TMAO (μmol/L)	1.16 (0.92–1.24)	1.22 (0.78–1.87)	1.00 (0.84–2.96)	0.93	0.38	**0.033**
IGF-1 (ng/mL)	129 (84–176)	148 (103–195)	184 (145–235)	1.00	**0.030**	0.24
Insulin (μU/mL)	13.4 (10.6–29.8)	16.0 (8.5–32.2)	10.0 (7.2–13.2)	1.00	**0.044**	**0.026**
hsCRP (mg/L)	3.65 (2.26–9.90)	1.31 (0.54–3.57)	1.54 (0.84–2.96)	**<0.001**	**0.005**	1.00
PSP (mg/L)	2.39 (2.25–2.83)	2.55 (2.20–2.79)	2.89 (2.67–3.44)	1.00	**0.008**	**0.020**

Table legend: IMA—ischemia modified albumin, AOPP—advanced oxidation protein products, SH groups—thiol groups, TAS—total antioxidant status, TMAO—trimethylamine N-oxide, IGF-1—insulin growth factor-1, hsCRP—high sensitive C-reactive protein, PSP—presepsin. Statistically significant differences are given in bold.

**Table 3 medicina-55-00585-t003:** Parameters (median with 25%–75%) of oxidative and inflammatory status in patients with myocardial infarction (T0) and after 3 months recovery after PTCA (T3) regarding to ejection fraction <50% and ≥50%.

Parameter (Unit)	T0			T3			EF < 50%	EF ≥ 50%
	EF < 50%	EF ≥ 50%	*p*	EF < 50%	EF ≥ 50%	*p*	T0 vs. T3	T0 vs. T3
	*n* = 10	*n* = 20		*n* = 10	*n* = 20		*n* = 10	*n* = 20
IMA (ABSU/g)	52.0 (42.5–80.3)	37.6 (32.0–46.6)	**0.007**	57.3 (42.5–80.3)	46.3 (40.0–51.9)	0.344	0.799	**0.030**
AOPP (μmol/L)	105.3 (78.9–125.6)	286.6 (181.1–400.5)	**0.015**	283.2 (141.1–315.1)	266.6 (181.4–397.0)	0.982	**0.017**	0.708
SH groups (mmol/L)	0.41 (0.34–0.45)	0.43 (037–0.46)	0.552	0.42 (0.20–0.50)	0.39 (0.26–0.46)	0.982	0.508	0.433
TAS (μmol/L)	17.5 (17.0–17.8)	16.3 (15.3–18.1)	0.379	17.1 (16.4–17.8)	16.5 (15.3–17.9)	0.135	0.386	0.550
TMAO (μmol/L)	1.13 (0.92–1.38)	1.22 (0.81–1.27)	0.758	1.74 (1.17–3.10)	1.03 (0.61–1.71)	**0.031**	0.093	0.970
IGF-1 (ng/mL)	107.4 (55.9–151.2)	154.2 (95.2–261.0)	0.056	117.4 (77.6–180.1)	153.9 (132.2–213.3)	0.209	**0.047**	0.526
Insulin (μU/mL)	13.4 (11.2–18.3)	13.9 (10.5–36.4)	0.644	18.7 (8.5–29.8)	18.0 (9.8–35.1)	0.947	0.169	0.970
hsCRP (mg/L)	11.00 (2.90–29.00)	3.17 (1.20–7.55)	**0.045**	1.68 (0.59–2.19)	1.18 (0.52–3.63) 2.38 (2.30–2.79)	0.741	**0.012**	**0.015**
PSP (mg/L)	2.41 (1.89–3.43)	2.38 (2.30–2.79)	0.982	2.21 (1.95–2.54)	2.59 (2.15–2.77)	0.159	0.284	0.794

Table legend: IMA—ischemia modified albumin, AOPP—advanced oxidation protein products, SH groups—thiol groups, TAS—total antioxidant status, TMAO—trimethylamine N-oxide, IGF-1—insulin growth factor-1, hsCRP—high sensitive C-reactive protein, PSP—presepsin, LV mass—Left ventricular mass, EF—ejection fraction. Statistically significant differences are given in bold.

**Table 4 medicina-55-00585-t004:** Results of the correlation analysis between parameters of oxidative-inflammatory status in MI patients (T0—baseline, T3—3-month follow-up) and control subjects (C).

Variables	r and *p*-Values
T0	
IMA vs. IGF-1	r = −0.376, *p* = 0.041
IMA vs. hsCRP	r = 0.466, *p* = 0.009
TAS vs. AOPP	r = −0.373, *p* = 0.042
TAS vs. SH groups	r = 0.409, *p* = 0.025
TAS vs. IGF	r = −0.375, *p* = 0.041
TMAO vs. hsCRP	r = 0.371, *p* = 0.044
TMAO vs. PSP	r = 0.381, *p* = 0.038
TMAO vs. insulin	r = 0.374, *p* = 0.042
T3	
TMAO vs. glucose	r = 0.374, *p* = 0.042
C	
PSP vs. hsCRP	r = 0.356, *p* = 0.039

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
