# Peer review of "Parameters of Oxidative and Inflammatory Status in a Three-Month Observation of Patients with Acute Myocardial Infarction Undergoing Coronary Angioplasty—A Preliminary Study"

_medicina, 2019, doi:10.3390/medicina55090585_

Round 1
Reviewer 1 Report
In this study the authors investigated oxidative and inflammatory parameters, which can be useful in myocardial infarction patients after percutaneous transluminal coronary angioplasty.
The results show that most of parameters changed between patients and control group, especially men subjects.
Methods of analysis and results are fully written in the manuscript. Oxidative and inflammatory markers are being presented in the discussion and the data are being compared with other authors‘ data.
Although, analysing conclusions there are missing clear and informative message to the readers.
So, I ask authors to answer my comment and add to conclussions, substantiating the results:
1. The conclusions should better represent the results.
Author Response
Reviewer 1.
1) The conclusions should better represent the results.
We rewritten the conclusions to made them clearly legible (page 10, lines 327-335).
Reviewer 2 Report
In this manuscript, Ewa et al. aimed to monitor the conditions of the patients with coronary angioplasty surgery by evaluate a series of oxidative and inflammatory factors. This is a single institutional study. While the experiment design was straightforward, and discovered some factors could serve as candidate parameters to evaluate the inflammatory and oxidative status in the patients, this reviewer concerns about the following issues:
1) Although the authors observed significance but minor alterations of some parameters. Those changes were not linked with the disease progression or recovery status which made this study a simple observation but did not give any further suggestion on management of post-surgery care.
2) The measurements were performed on the patients at two time points. But these measurements only performed once on the control group. As the values of all the parameters except AOPP and hsCRP were very close in patients and control group (Differences are less than 1 fold). The control should be included each time when the patients were checked to minimize the possibility that differences among these groups were caused by variabilities between different measurements.
3) Investigation with larger group and longer time of follow-up might made the result sound.
4) The author should clearly indicate the number of each gender in each group used in the experiment. As shown in table 1, the male sex percentage was 63% in the control group. So how many males in this group?
Author Response
Reviewer 2.
1) Although the authors observed significance but minor alterations of some parameters. Those changes were not linked with the disease progression or recovery status which made this study a simple observation but did not give any further suggestion on management of post-surgery care.
On the basis of the newest literature data and our own experience, we preselected to this study a set of parameters that have never been determined simultaneously during myocardial infarction (MI) and never been compared after 3-month convalescence. We expected more distinctive changes between analyzed groups, however our assumptions were not confirmed. The more significant changes are generally observed between patients and control group, not between time of MI and 3-months later, which we were most interested in. We did not find any parameter which differ significantly among all groups. The most pronounced changes were noticed for hsCRP, SH groups and AOPP. The first one is well-known biomarker in CVD with no signs of novelty. Nonetheless, changes in SH groups and AOPP level are not so obvious, which seems to us worth presenting. Moreover, additional statistical analysis in subgroups of patients divided according to the left ventricular function revealed some new observations, which was added to the manuscript body.
As the reviewer point out, those changes were not directly linked with the disease progression or recovery status. However, examined by us parameters may somehow reflect biochemical status and this observation per se seems to us interesting. It should be taken into account, that patients included to this study were basicly at the same state – they all survived and upon control visit after 3 months they did not complain angina pain. But still we observed significant changes in most parameters in comparison to control group. We added this comment to the “Discussion” section (page 10, lines 321-326).
2) The measurements were performed on the patients at two time points. But these measurements only performed once on the control group. As the values of all the parameters except AOPP and hsCRP were very close in patients and control group (differences are less than 1 fold). The control should be included each time when the patients were checked to minimize the possibility that differences among these groups were caused by variabilities between different measurements.
We agree with the reviewer that additional check point for control group may bring some valuable observations and that’s how we planned the experiment. Unfortunately, the majority of the control group did not return to repeat blood collection and we have incomplete data, which were not presented. However, in few measured samples, individual intra-variability was not greater than 10%.
The aim of our preliminary study was to evaluate a set of markers to characterize oxidative-inflammatory status in MI patients during incidence and after coronary angioplasty. If look at the results in patients group, apart from hsCRP (which is validated parameter, with known individual intra-variability), the more significant difference between measurements in T0 and T3 group was observed for AOPP and even though it was not significant. The control subjects in this study serves mainly as reference group. They not have any cardiovascular diseases and their general health state was stable. Following this course of thought, even if any changes in the analyzed parameters occur, they were not caused by infarction or PTCA. However, in order to minimize the influence of possible other factors on the test results in the control group, we used the same results from the control group to compare with patients group at both measuring points (T0 and T3).
However, to explain the reviewer doubts, the variability of measurements for each analytical method was been either established in our laboratory (according to “Guideline for the Quality Assurance of Laboratory Medical Examinations”, https://www.egms.de/static/de/journals/lab/2015-6/lab000018.shtml) or provided by the ELISA kit manufacturer. For the most important parameters intra- and inter-assay CV were as follows: AOPP 5.9% and 10.5%, SH 6.2% and 10,5%, TMAO 1.77% and 6.58% (as we pointed out in “Method section” page 3, line 123), presepsin 7% and 11%, respectively (according to manufacturer protocol).
3) Investigation with larger group and longer time of follow-up might made the result sound.
We agree with the reviewer that our study would be more relevant with larger group and longer time of follow-up as we also mentioned in the “Discussion” section” (page 10, lines 331-335). These is preliminary study and we consider that carrying out research in this direction is worth continuing, which was mentioned. We have already planned to extend the investigations, but to get the grant funding, it would be desirable to have any preliminary results published, which we hope for.
4) The author should clearly indicate the number of each gender in each group used in the experiment. As shown in table 1, the male sex percentage was 63% in the control group. So how many males in this group?
21 patients and 19 controls were males. It was corrected in the table 1 (page 4).
Reviewer 3 Report
I suggest comparing the OS markers with the extension of myocardial damage (troponin, left ventricle EF). Please add type of myocardial infarction- STEMI or NSTEMI; did patients receive stent or not, is it 1-, 2- or 3- vessel disease. Add medications that patients received at discharge. It would be nice to compare biomarkers that authors investigated in patients with STEMI/NSTEMI, patients with 1, 2- or 3 -vessel disease.
Discussion should be improved. Study limitations should be added.
Author Response
Reviewer 3.
1) I suggest comparing the OS markers with the extension of myocardial damage (troponin, left ventricle EF).
We appreciate the opportunity to include additional results. No significant correlations was revealed with examined parameters and LV EF, but deepening analysis in direction pointed out by the reviewer bring some new observations. We added a table with data comparing patients with normal (≥50%) and impaired (<50%) ejection fraction to the manuscript body (it is currently numbered as Table 3). Since oxidative stress and inflammation contribute significantly to progression of heart failure, parameters of oxidative and inflammatory balance were expected to be more pronounced in patients with low EF. Among all parameters IMA, AOPP and hsCRP differed significantly between groups in T0 point, and TMAO in T3 point.
2) Please add type of myocardial infarction- STEMI or NSTEMI; did patients receive stent or not, is it 1, 2- or 3- vessel disease. Add medications that patients received at discharge.
We added all missing information to “Study population” section (page 2, lines 83-89). 17 patients were classified as STEMI type, and 13 as NSTEMI. These were patients mainly with 1-vessel disease (24 subjects), only 4 were with 2-vessels. All of them received stents. The details of the received medications have been supplemented.
3) It would be nice to compare biomarkers that authors investigated in patients with STEMI/NSTEMI, patients with 1, 2- or 3 -vessel disease.
We thank the reviewer for these suggestions. Comparing patients with STEMI and NSTEMI we did not reveal any significant differences in parameters of oxidative-inflammatory balance (Table III), so they seem to be uninfluenced by the MI type. Taking under account, that there is not much variations in these results, we did not added whole table to the manuscript body, but we entered a short commentary both in “Results” and in “Discussion” section and added this table to supplementary data (page 5, lines 159-160 and page 10, lines 310-311, page 11, line 329).
Round 2
Reviewer 2 Report
The authors have tried their best to answer all my questions.
Author Response
The authors have tried their best to answer all my questions.
Thank you, we have tried very hard to clarify any doubts.
Reviewer 3 Report
I suggest not mention the impact of PCI on oxidative and inflammatory status. I suggest focussing on ACUTE MI, since the blood was drawn before PCI at admission for T0 (correct if the blood was drawn at a certain time point after PCI). We can not measure the impact of PCI and MI separately on OS and inflammatory markers. Further, the control group did not undergo percutaneous coronary intervention. PCI should be mentioned as a contemporary therapy which indeed increases the intensity of OS, but its measurement was not the aim of this study.
Did patients receive statins, beta-blockers, beside antithrombotic therapy and ACE, after MI in secondary prevention? Were all patients underwent CV rehabilitation and what that included? This should be defined.
Parts Introduction and Discussion should be significantly shortened. We have many different biomarkers in a very complex situation at 2-time points. Try to simplify it. I would create predictive models with fewer biomarkers for example...
Author Response
I suggest not mention the impact of PCI on oxidative and inflammatory status. I suggest focusing on ACUTE MI, since the blood was drawn before PCI at admission for T0 (correct if the blood was drawn at a certain time point after PCI). We can not measure the impact of PCI and MI separately on OS and inflammatory markers. Further, the control group did not undergo percutaneous coronary intervention. PCI should be mentioned as a contemporary therapy which indeed increases the intensity of OS, but its measurement was not the aim of this study.The aim of our study was to evaluate a set of markers to characterize oxidative-inflammatory status in patients with myocardial infarction (MI), who then underwent coronary angioplasty. While in T0 time point (infarction incidence) measured parameters indeed reflected patients biochemical status caused by MI, we can’t exclude that in T3 (after 3-monts follow-up) these parameters are affected not only by MI, but also by PTCA. There are many literature data indicating that PTCA “induces disturbances in the biochemical balance of the organism, e.g. it exacerbates the inflammatory response, generates the production of free radicals, and also initiates the activation of many signaling pathways” as we mentioned firstly in “Introduction” section. That is why we can’t ignore the possible impact of PTCA on the measured parameters. As we discussed: “The full improvement of vascular functions after the PTCA occurs progressively over a period of time. (…) According to scientific reports, the majority of improvements after PTCA occur in the first 3 months. This period includes also restoring of the disturbed biochemical balance, therein amplification of proinflammatory mechanisms, coagulation imbalance, endothelial dysfunction and oxidative-antioxidative disturbance”. However, until now it wasn’t known how examined by us parameters changed over the 3 months period of time. Of course, it would be advisable to measure these parameters also immediately after PTCA, but we don't have such possibility now. We are planning to add an additional measurement point in the further studies with larger group of participants. In this preliminary study we want to choose some of the most promising parameters we would like to focus on in the future.
However, the reviewer's doubts made us aware that the readers' attention may be directed at PTCA, which wasn’t our intention. Therefore, we to rewrite the manuscript again and remove some fragments about the PTCA.
Did patients receive statins, beta-blockers, beside antithrombotic therapy and ACE, after MI in secondary prevention? Were all patients underwent CV rehabilitation and what that included? This should be defined.
We thank the reviewer for his vigilance. All patients were treated with beta-blockers after PTCA, which we forgot to add in the previous correction. Now, the pharmacological recommendations are finally supplemented and we ask for the reviewer's understanding of our mistake. Moreover, all patients received statins before admission to the hospital and this the treatment was maintained after the procedure. According early post-hospital rehabilitation (lasting 8-12 weeks), it included low-intensity resistance exercises 3 times a week, 10-15 repetitions each and/or stationary bike workout (starting from 15 minutes and gradually extending), as well as individually selected general fitness exercises (e.g. fast march). This clinically important information is now included in the “Study population” subsection.
Parts Introduction and Discussion should be significantly shortened. We have many different biomarkers in a very complex situation at 2-time points. Try to simplify it. I would create predictive models with fewer biomarkers for example.
We have made every effort to shorten and simplify the manuscript. However, as reviewer noticed, the situation is very complex. We have different parameters, 2 time points and relatively small group of participants, which allows us to perform only basic statistical analysis. We also decided for discriminate analysis, which allowed us to determine which of the analyzed variables discriminate subjects between two or more naturally occurring groups (in this study into T0, T3 and control groups). Typically, it is a method for predicting some level of a one-way classification based on known values of the responses. In our preliminary study, this analysis was a tool for choosing the most promising and strongest parameters describing the observed phenomenon. Knowing the current limitations of this study, we assume the possibility of creating reliable predictive models only after obtaining results from a much larger group of participants.
Round 3
Reviewer 3 Report
Authors significantly improved the paper.
Please correct in the Abstract section - "patients who underwent MI"....WITH ...."patients with MI who underwent PCI", etc
Please add P value in Table 1 for PRESENCE OF DM
Do correct typos.
Author Response
Authors significantly improved the paper.
Thank you for appreciating our efforts and we are very pleased that we were able to meet the reviewer's requirements.
Please correct in the Abstract section - "patients who underwent MI"....WITH ...."patients with MI who underwent PCI", etc
Corrected (see manuscript with track changes mode)
Please add P value in Table 1 for PRESENCE OF DM
Added (chi-squared test with Yates correction was applied - suitable in groups when the number of cases is less than 5).
